# Quality of Care in Public County Hospitals: A Cross-Sectional Study for Stroke, Pneumonia, and Heart Failure Care in Eastern China

**DOI:** 10.3390/ijerph19159144

**Published:** 2022-07-27

**Authors:** Quan Wang, Li Yang, Jialin Chen, Xi Tu, Qiang Sun, Hui Li

**Affiliations:** 1Center for Health Management and Policy Research, School of Public Health, Cheeloo College of Medicine, Shandong University, Jinan 250012, China; quann.wang@mail.utoronto.ca (Q.W.); qiangs@sdu.edu.cn (Q.S.); 2National Health Commission (NHC) Key Lab of Health Economics and Policy Research, Shandong University, Jinan 250012, China; 3School of Public Health, Peking University, Beijing 100191, China; lyang@bjmu.edu.cn; 4Shandong Provincial Hospital for Skin Diseases & Shandong Provincial Institute of Dermatology and Venerology, Shandong Academy of Medical Sciences, Jinan 250022, China; 202016365@mail.sdu.edu.cn; 5School of Health Policy and Management, Chinese Academy of Medical Sciences & Peking Union Medical College, Beijing 100005, China; tuxi@student.pumc.edu.cn

**Keywords:** stroke, pneumonia, heart failure, quality of care, public hospitals, China

## Abstract

There are very few studies about the quality of care (QoC) in Chinese county hospitals. Using 7, 6, and 6 standard operations from clinical pathways as the process indicators, we evaluated the quality of stroke, pneumonia, and heart failure care, respectively. We also conducted chi-squared tests to detect differences of quality between selected counties or hospitals. We extracted relevant information from medical records of 421 stroke cases, 329 pneumonia cases, and 341 heart failure cases, which were sampled from 6 county hospitals in 3 counties of eastern China. The average proportion of recommended care delivered included stroke, pneumonia, and heart failure patients at 55.36%, 41.64%, and 49.56%, respectively. Great variation of QoC was detected not only across selected counties but between comprehensive county hospitals and traditional Chinese medicine county hospitals. We deny the widely-accepted assumptions that poor QoC should be blamed on defectively-equipped facilities or medicine and overwhelmed care providers. Instead, we speculate the low qualifications of medical workers, failed clinical knowledge translation, incorrect diagnosis, and a lack of electronic systems could be the reasons behind poor QoC. It is high time for China to put QoC as the national health priority.

## 1. Introduction

An increasing number of policy makers have accepted the idea of universal health coverage (UHC), which pledges to provide all people with essential health care services at an affordable price. It has long been recognized that quality of care (QoC) is an essential link between healthcare and health outcomes [1]. Therefore, as suggested by many experts, without quality, universal health coverage remains an empty promise [2,3]. A 2018 study led by The Lancet Global Health Commission concluded that low-income and middle-income countries (LMICs) lost more than 8 million people per year due to a low-quality healthcare system [4]. In addition, substandard quality of care also exerts considerable cost implications on the healthcare system and individuals. A systematic report issued by the World Health Organization (WHO), OECD, and the World Bank Group in 2018 pointed out that health services of poor quality disproportionately affected vulnerable groups in society and resulted in trillions of dollars lost [5].

Since 2010, public hospitals in China provide over 60% of the country’s inpatient services and more than 30% of outpatient services, and the numbers are rising year by year [6]. Standing at the top of a three-tiered service delivery system (county hospital, township hospital, and village clinic) in rural areas of China, county hospitals are designed to provide essential medical services that meet the health needs of more than 900 million people, approximately 70% of the Chinese population [7,8]. Generally, there are 3 types of county hospitals: the general county hospital, the traditional Chinese medicine (TCM) hospital, and the maternal and child healthcare hospital. In 2014, the second phase of China’s health care system reform when the transformation of resources into effective services became the national health priority [9], the central government issued a specific regulation to guide the development of county hospitals. Local governments were required to prioritize the development of at least 1 or 2 county hospitals within their jurisdictions [10]. Therefore, there is at least 1 public general county hospital and 1 public TCM county hospital in most counties of China. In 2015, there were 13,074 county hospitals in China, and, 5 years later, the number reached 16,804, an increase of 28.53%. In 2020, county hospitals resolve 1.16 billion outpatient visits and 80.65 million inpatient admissions [11]. However, there is limited evidence of improvements in QoC and health outcomes in China over the last decade [12], especially for county hospitals. According to Donabedian, a framework consisting of 3 dimensions was recommended to evaluate quality: (1) structure, or the characteristics of a health care setting; (2) process, or what is done to patients; and (3) outcomes, or how patients fare after health care interventions [13,14]. Currently, due to lack of accessible data, almost all studies about QoC of Chinese hospitals are focused on structures or outcomes [15,16].

Without enough evidence, policy makers cannot predict county hospitals’ behavioral responses to policy options and therefore design policies that would maximize the QoC. Therefore, the primary objective of this study is to evaluate the QoC of county hospitals in China by using process indicators. We selected 3 common diseases: stroke, pneumonia, and heart failure. Stroke is the second leading cause of death in the world and the leading cause of death in China [17]. As estimated by Ma et al., 28.8 million strokes occurred in China in 2019, which caused 2.19 million deaths and 45.9 million disability-adjusted life-years (DALYs) [18]. The prevalence of stroke in China was 2022 per 100,000 population in 2019, higher than Europe (1870 per 100,000 population in 2017) [18,19]. Pneumonia is a leading infectious cause of hospitalization and death among adults [20]. According to Hu et al., the hospitalization rate for pneumonia was 8.4 admissions per 1000 person-years between 2009 and 2017 and increased year by year [21]. Despite the fact that the length of hospital stay slightly decreased during the same time span, the hospitalization burden of pneumonia in Chinese adults increased [21]. A population-based study of about 50 million individuals shows the overall prevalence of heart failure was 1.18% in China in 2017 [22]. Compared with developed countries (2.2% in the U.S. and Sweden, 4% in Germany, 1.6% in UK) [23,24], the prevalence of heart failure in China stays at relatively low levels. The inpatient and outpatient mean cost per capita was $4406.8 and $892.3, which placed a considerable burden on health systems in China [22]. Stroke and pneumonia also rank as top 2 diseases for inpatients in Chinese county hospitals [11]. In addition, clinical pathways for these 3 diseases exist in China, and their standard operations are fully covered by the services of the county hospital.

This study is a part of the project Evaluating Health Care Quality for China’s County Level Hospitals, which is funded by the China Medical Board (CMB). The data of this study are from Shandong Province and can represent the QoC in the eastern part of China. The results can be used to inform the government on its county hospital policies and serve as a baseline to assess policy interventions.

## 2. Methods

### 2.1. Sampling and Data Source

We randomly selected 3 counties in Shandong Province. In each county, we chose the largest comprehensive county hospital and the largest TCM county hospital, since they provided most of the care services for the local residences.

From all inpatients whose first diagnosis at discharge was pneumonia (ICD-10: J15) or heart failure (ICD-10: I50) in 2016, we randomly selected 5 cases in every included hospital per month by using a systematic sampling method. Considering the difficulty of stroke diagnosis and the resultant errors, we randomly chose 6 cases whose first diagnosis at discharge was stroke (ICD-10: I63) in 2016 in every included hospital per month. The random selection process was conducted in September of 2017. A total of 360 pneumonia cases, 432 stroke cases, and 360 heart failure cases were included in this study (Check Appendix A for detailed ICD-10 codes).

### 2.2. Data Extraction

No shared information system existed among the 6 selected hospitals, which left us with no choice but to manually take photos of all included medical records manually. Specific data extraction charts were developed for stroke, pneumonia, and heart failure cases. Discharge information was collected through the medical records, including patients’ age, gender, payment source, admission route, severity, history of drug allergy, and anamnesis. All treatment information related to QoC indicators was also obtained. No personal or identifiable information was extracted or recorded in this study. Two reviewers (JC and HL) conducted data extraction independently, and the conflict between the two reviewers’ results was discussed and eventually resolved with another author’s help (QS). After reviewing samples of medical records from the 6 selected hospitals, 11 stroke cases, 31 pneumonia cases, and 19 heart failure cases were excluded due to important information being missing. Ultimately, we included 421 stroke cases, 329 pneumonia cases, and 341 heart failure cases in the analysis.

### 2.3. Quality of Care Indicators

Considering the clinical guideline recommendations from the Chinese Medical Association (CMA) and National Quality Forum (NQF) [25,26,27,28], we utilized the following indicators to evaluate the QoC:

#### 2.3.1. Indicators for Stroke Care

(1)Anticoagulant drugs at arrival: Percentage of stroke patients that received anticoagulant drugs within 24 h after hospital arrival.(2)Aspirin at arrival: Percentage of stroke patients that received aspirin within 24 h after hospital arrival.(3)Prescriptions at discharge: Percentage of stroke patients that got prescriptions at hospital discharge.(4)Aspirin at discharge: Percentage of stroke patients that were prescribed aspirin at hospital discharge.(5)Statins at discharge: Percentage of stroke patients that were prescribed statin at hospital discharge.(6)Antihypertensive drugs at discharge: Percentage of stroke patients that were prescribed antihypertensive drugs.(7)Health education: Percentage of smoking stroke patients that received smoking cessation education.

#### 2.3.2. Indicators for Pneumonia Care

(1)Evaluation of blood oxygenation: Percentage of pneumonia patients that received blood oxygenation evaluation.(2)Antibiotics within 6 h: Percentage of pneumonia patients that took antibiotics within 6 h after arrival.(3)Selection of antibiotics: Percentage of pneumonia patients that took antibiotics as suggested by clinical guidelines.(4)Blood culture: Percentage of pneumonia patients that underwent blood cultures before antibiotic treatment.(5)Sputum culture: Percentage of pneumonia patients that underwent sputum cultures before antibiotic treatment.(6)Health education: Percentage of smoking stroke patients that received smoking cessation education.

#### 2.3.3. Indicators for Heart Failure Care

(1)Evaluation of left ventricular systolic (LVS) function: Percentage of heart failure patients that received LVS function evaluation.(2)Angiotensin converting enzyme inhibitor (ACEI) at arrival: Percentage of heart failure patients that received ACEI.(3)Angiotensin receptor blocker (ARB) at arrival: Percentage of heart failure patients that received ARB.(4)Prescriptions at discharge: Percentage of heart failure patients that were prescribed aspirin, ARB, or statin at hospital discharge.(5)Health education on diet: Percentage of heart failure patients that received diet education.(6)Medical advice on rest: Percentage of heart failure patients that received medical advice to take more rest than usual.(7)Medical advice on follow-up visit: Percentage of heart failure patients that received medical advice on follow-up visit.

All the indicators were selected based on care recommended by clinical guidelines from the CMA and NQF. Except for smoking cessation education, the denominator was the number of patients included, and the numerator was the number of patients who received corresponding care. As for smoking cessation education, the denominator was the number of patients who were smoking, and the numerator was number of patients who received smoking cessation education.

### 2.4. Analysis Method

We utilized sample descriptive statistics to measure the characteristics of the included samples and the QoC of hospitals in each county. The chi-squared tests were also adapted to detect differences in quality between counties or hospitals. All data analysis was conducted with SPSS 21.0 (IBM Corporation, Armonk, NY, USA).

## 3. Results

Three counties were selected: Linyi, Zoucheng, and Qingzhou, where 6 hospitals in total were included in this study. The basic characteristics of these hospitals are shown in Table 1. Generally, the comprehensive county hospitals were larger than the TCM county hospitals, with more staff, departments, and hospital beds. As regulated by the local administration, the proportion for medical workers and managers should be above 80% and 10%, respectively [29]. All 6 hospitals met this requirement for medical workers. However, the managers were quite low, even staying at the level of 1.39%.

### 3.1. Stroke

#### 3.1.1. Characteristics of Included Stroke Patients

Of all 421 stroke patients (Table 2), the average age was 68, and 250 (59.38%) of them were male. Most were covered by urban and rural resident basic medical insurance (URRBMI), 395 (93.82%) were admitted from the outpatient department, and only 6 were recorded with a history of drug allergy. Regarding severity, 176 (41.81%) patients were diagnosed with mild stroke, followed by moderate stroke (25.42%) and severe stroke (22.33%).

#### 3.1.2. QoC of Stroke Care

The results of the QoC indicators are shown in Table 3 and Table 4. Anticoagulant drugs within 24 h were administered to 390 (92.64%) stroke patients, and 338 (80.29%) received aspirin. When patients were discharged from the hospitals, 301 (71.50%) of them received prescriptions. The percentage of patients in Linyi, Zoucheng, and Qingzhou who received anticoagulant drugs at arrival was 97.22%, 85.42%, and 95.49%, respectively; for prescriptions at discharge, the numbers were 63.19%, 78.47%, and 72.93%. The results suggested poor quality of stroke care in selected hospitals, except for anticoagulant drugs at arrival, and all the indicators were below 90%. The care at arrival was better than discharge care. Even the lowest index for the former, aspirin at arrival, was higher than the highest index for the latter. Health education was the worst indicator, and only 5 of all 56 smoking patients revived smoking cessation education. We also detected significant variations in most indicators among the 3 counties and 2 kinds of hospitals.

### 3.2. Pneumonia

#### 3.2.1. Characteristics of Included Pneumonia Patients

We extracted information from 329 medical records. The average age for included pneumonia patients was 46.34 years, and 183 (55.62%) of them were male. Most (76.29%) of them were covered by URRBMI, followed by those covered by urban employee basic medical insurance (UEBMI). As for the admission route, 315 (96.05%) were hospitalized from the outpatient department (OD). Check Table 5 for detailed information.

#### 3.2.2. QoC of Pneumonia Care

On average, only 34.04%, 3.65%, and 32.83% of the included pneumonia patients received blood oxygenation evaluation, blood culture, and sputum culture, respectively. None of the 51 smoking patients received smoking cessation education when they were discharged. In looking at antibiotics, the indicators for antibiotics within 6 h and suggested antibiotics stayed at more than 85%. Given the level of the indicies, it is unsurprising that the quality of pneumonia care was poor, especially for diagnoses. The QoC in comprehensive county hospitals was better than that in TCM county hospitals. Except for health education, which stayed at 0%, all the indicators in the former were better or stayed at the same level. Quality differences were detected among the 3 counties for blood oxygenation evaluation and blood culture. In addition, there were considerable variations between comprehensive hospitals and TCM hospitals for blood oxygenation evaluation, blood culture, and sputum culture. Outcomes of each quality measure are reported in Table 6 and Table 7.

### 3.3. Heart Failure

#### 3.3.1. Characteristics of Included Heart Failure Patients

Of all 341 heart failure patients, more than half (64.22%) were female, and the average age was 68.38 years (Table 8). For payment, 269 (78.89) of them were covered by URRBMI, followed by those covered by UEBMI (16.42%). The outpatient department was the leading route for admission, by which 291 (85.34%) of the included patients were hospitalized. None of the patients was diagnosed with mild heart failure, and most (81.23%) of them had moderate heart failure.

#### 3.3.2. QoC of Heart Failure Care

The county hospitals only provided 202 (59.24%) patients with left ventricular systolic (LVS) function evaluation, and the proportions of patients who received angiotensin converting enzyme inhibitor (ACEI) and angiotensin receptor blocker (ARB) at arrival were incredibly low, less than 10%. The care at discharge was better than the care at arrival. Approximately 78.59% of patients got prescriptions like aspirin, statins, or ARBs when discharged from the hospital. Also of note was that 42.82%, 72.43%, and 79.47% of heart failure patients received health education on diet, rest, and follow-up visits, respectively. Except for LVS evaluation and ARB at arrival, the quality of care shared great variation among Linyi, Zoucheng, and Qingzhou. Compared with quality differences among the 3 counties, the difference in QoC between comprehensive hospitals and TCM hospitals was smaller. Heterogeneity was only detected for LSV evaluation and ARB at arrival. Check Table 9 and Table 10 for detailed information.

## 4. Discussion

In this study of 6 hospitals from 3 counties in eastern China, we find that widespread gaps between reality and expectation in QoC still exist. The 3 diseases in this study are rather common for county hospitals in China: according to the China Health Statistics Yearbook, the proportions of inpatients in the county hospital diagnosed with stroke, pneumonia, and heart failure were 4.83%, 5.65%, and 0.73%, respectively [11]. Also of note is that the stroke and pneumonia are the top 2 diseases for inpatients in Chinese county hospitals [11]. In addition, the clinical pathways of these diseases are clear, and all what county hospitals need to act on is the timely and correct implementation of standard operations. However, the average proportions of recommended care delivered for stroke, pneumonia, and heart failure patients were merely 55.36%, 41.64%, and 49.56%, respectively. Considering the principle of best practice, we believe there is still great room for quality improvement.

As part of our findings, significant differences in QoC were identified not only among the 3 selected counties, but also between the comprehensive county hospitals and TCM county hospitals. Overall, the QoC in comprehensive county hospitals seems better than that of TCM county hospitals, and the drug-related indicators are better than the test-related indicators. For stroke and pneumonia, the QoC at arrival is better than it is at discharge, whereas for heart failure, the reverse applies. We speculate that this may be caused by the course of the disease. Compared with stroke and pneumonia, heart failure is a chronic disease, and treatment during a patient’s daily life is much more important than during hospitalization.

We believe these findings suggest that it is high time for China to put QoC as a health priority, especially the QoC in county hospitals. Just as many other counties, during the past decade, the Chinese government and society made access the national health priority [30]. As a consequence, the monitoring indicators and evaluation criteria of the health care system evaluate structures, like inputs of medicines and equipment, but do not capture many of the processes and outcomes that matter most to people, like correct diagnosis, timely treatment, and user experience [4]. Motivated by access-oriented evaluation criteria, China has expanded the size of doctors, nurses, hospital beds, and even medical institutes rapidly during the last 10 years [9,31]. Unfortunately, the QoC seems to have failed to follow the trend. In addition, due to deficient payment methods and financing systems, hospitals in economically developed counties usually get more income than hospitals in economically underdeveloped counties [32], which could cause great quality differences like those found in this study. From equity to quality, we call for a novel health development route to achieve better health outcomes and equity for all the Chinese population.

Another significant question is what leads to the poor QoC of Chinese county hospitals? Many people blame defectively-equipped facilities and medicines. However, our study does not support this assumption. The QoC indicators we employed in this study only involve some quite common drugs, like aspirin, ACEI, ARB, etc., and simple tests, like blood oxygenation assessment and LVS function evaluation. All of these medications and tests are fully accessible in county hospitals, and are financially affordable for patients, more than 95% of whom are covered by UEBMI or URRBMI. Although the indicators for health education or medical advice at discharge require almost no resources for doctors or hospitals, they are still the worst among all QoC indicators, staying as low as approximately 10%. We also deny the assumption that poor quality may be due to overwhelmed care providers. According to the China Health Statistics Yearbook, doctors in the county hospitals of the Shandong Province only treat 5.8 outpatients and provide 2.5 bed days of service per day, on average [11].

Based on these findings, we propose 5 possible reasons. First, low qualifications. In the selected hospitals, only 10% of doctors and nurses held a master’s degree or above, whereas 43% did not even have a bachelor’s degree. The finding that TCM county hospitals, whose doctors and nurses had inferior qualifications compared with comprehensive hospitals, got worse QoC than comprehensive hospitals might verify this point. Second, as suggested by Das, clinical knowledge often fails to translate into clinical practice [3,33]. For instance, we discussed with several doctors in selected hospitals about smoking cessation education. Some of them told us they would advise patients against smoking at discharge, while they just did not write this on medical records. Third, we utilized the diagnosis at discharge as one of the inclusion criteria. There might be some patients who were diagnosed incorrectly at arrival, which delayed their receiving correct care services. Fourth, too few managers. The government suggested that the value for the proportion of managers in the county hospital was 10%, which none of the selected hospitals achieved. The lack of staff for administrative work may lead to extra burdens on medical staff and impede the delivery of high-quality care. Finally, not all hospitals were equipped with electronic systems. For 3 TCM hospitals, 2 and 3 of them, respectively, lack laboratory information systems (LIS) and picture archiving and communication system (PACS); therefore, some treatments or tests may fail to be imported into medical records. During the data extraction process, we found a significant portion of medical records missed some information, like cost, hospital stay, etc., which led to the exclusion of 48 records in total.

Nevertheless, it is encouraging that the quality improvement plan (QIP) does not require extensive investment, since we think the poor QoC should not be attributed to the lack of medications or medical equipment. First, we recommend enhancing training for medical workers in county hospitals, especially in-service training. It is impractical to improve the qualifications of doctors and nurses in a short time, and this could be fixed by in-service training. The training guideline should be gently applied and tailored for county settings, and it should encourage trainees to translate knowledge into clinical practice. Clinical pathways (CPWs) also can be a powerful tool to guide evidence-based healthcare in Chinese county hospitals, which aim to translate clinical practice guideline recommendations into clinical processes of care within the unique culture and environment of a healthcare institution [34]. Second, refine the defective payment method and financing system. Currently, the predominant payment form in China is still fee for service, which leads to a relatively lower income of workers in lower-level medical institutes [35]. Therefore, doctors and nurses with high qualifications tend to choose higher-level hospitals than county hospitals. In addition, due to limited financial resources, hospitals have to reduce the size of managers. The new payment method and financing system should be quality- and value-oriented and flatten the income gradient among all levels of medical institutes. Third, establish accountability for quality. Accountability is highly important to the success of QIP [36]. Currently, most monitoring indicators for the healthcare system in China are focused on access rather than quality, and the systematic set of quality indicators about the management of medical processes is still missing [15]. Although China’s central government issues a comprehensive report on the national QoC every year, its influence is still obscure. In addition, a well-functioning comprehensive electronic medical system is a prerequisite for accountability. Without complete documented data, health authorities cannot identify the problems from massive medical records, and the solution is consequently impossible. Generally, training could improve QoC, especially the QoC for common diseases, in a relatively short time without large-scale investment. However, it cannot solve the fundamental problem. In contrast, the latter 2 recommendations would not take effect in the short term. From a longterm perspective, the quality- and value-oriented payments would help county hospitals recruit better-educated medical workers. Accountability can help authorities transfer tremendous amounts of energy and enthusiasm for quality improvement into concrete actions.

Although, as mentioned above, studies on the process dimension of QoC in China are very few, we still identified 2 possible relevant studies. Lin conducted a study on the quality of pneumonia care of 24 hospitals across southwestern China and utilized 3 QoC indicators: (1) oxygenation assessment, (2) antibiotic treatment, and (3) first antibiotic treatment within 6 h after admission [37], of which 2 are the same as our study. As estimated by Lin, 2.3% of the included pneumonia patients received blood oxygenation assessment, which in our study was 34.04%, and 91.1% received first antibiotic treatment within 6 h after admission [37], which in our study was 88.75%. Although the latter indicator in southwestern China seems a bit better than that in eastern China, the former indicator suggests great differences between the 2 places. We think the difference hints at great heterogeneity of QoC among different parts of China, namely from province to province. However, Lin’s study did not provide information about variation in QoC among the included hospitals, and, thus, we cannot infer whether heterogeneity of QoC exists within southwestern China. Zhou conducted a similar study on the quality of acute myocardial infarction (AMI) care in Beijing, the capital of China [15]. The process indictors for AMI care and stroke care are similar, both of which consist of aspirin at arrival and statins at discharge. Compared with our study, the indicators for AMI care in Beijing are slightly better than the indicators for stroke care in eastern China.

Our study has several limitations. First, this study was only performed in Shandong Province, which cannot represent the overall QoC status of China. Second, the exclusion of 48 pieces of records may introduce selection and reporting bias into this study. The medical records from different hospitals followed different paradigms since there is no universal hospital information system (HIS) across the country. For instance, some included records applied ICD-10 codes, whereas some included records applied self-made coding systems. Therefore, it was very difficult for us to extract information from some hospitals even by taking photos. Third, the measurement of this study was fully based on documented records. Some care services might be taken but not recorded, such as the smoking education we mentioned above.

## 5. Conclusions

Our study finds there is still great room for improvement of stroke, pneumonia, and heart failure care in eastern China, and the QoC varies among different places and hospitals. QoC in comprehensive county hospitals seems better than TCM county hospitals, and the implementation of drug-related care services seems better than test-related care services. QIP should be conducted by county-setting tailored in-service training, new quality- and value-oriented payment, and accountability for quality.

## Figures and Tables

**Table 1 ijerph-19-09144-t001:** Characteristics of the included 6 hospitals.

	Linyi	Zoucheng	Qingzhou
**Comprehensive hospital**			
Clinical department	67	68	66
Staff	974	1851	1287
Medical worker *N (%)*	863 (88.60)	1595 (86.17)	1087 (84.46)
Doctor *N (%)*	329 (38.12)	527 (33.04)	390 (35.88)
Nurse *N (%)*	468 (54.23)	792 (49.66)	567 (52.16)
Manager	79 (8.11)	109 (5.89)	99 (7.69)
Other	32 (3.29)	147 (7.94)	101 (7.85)
Education *			
Postgraduate *N (%)*	39 (4.89)	139 (10.54)	163 (17.03)
Undergraduate *N (%)*	352 (44.17)	640 (48.52)	594 (62.07)
Junior college or polytechnic school *N (%)*	406 (50.94)	540 (40.94)	200 (20.90)
Hospital Bed	808	696	900
**TCM hospital**			
Clinical department	46	47	45
Staff	506	505	488
Medical worker *N (%)*	405 (80.04)	471 (93.27)	425 (87.09)
Doctor *N (%)*	139 (34.32)	159 (33.76)	135 (31.76)
Nurse *N (%)*	184 (45.43)	192 (40.76)	160 (37.65)
Manager	45 (7.89)	7 (1.39)	8 (1.64)
Other	56 (11.07)	27 (5.35)	55 (11.27)
Education *			
Postgraduate *N (%)*	13 (4.02)	30 (8.55)	20 (6.78)
Undergraduate *N (%)*	91 (28.17)	137 (39.03)	101 (34.24)
Junior college or polytechnic school *N (%)*	219 (67.80)	184 (52.42)	174 (58.98)
Hospital Bed	360	340	432

* Education: the most advanced education level of doctors and nurses.

**Table 2 ijerph-19-09144-t002:** Characteristics of included stroke patients (*N* = 421).

Variables		*N* (%)
*N*		421 (100.00)
Age	0−18	0 (0)
	19−40	7 (1.66)
	41−60	118 (28.03)
	61−80	240 (57.01)
	>80	56 (13.30)
Gender	Male	250 (59.38)
	Female	171 (40.62)
Payment	UEBMI	64 (15.20)
	URRBMI	344 (81.71)
	Fully covered by government	2 (0.48)
	OOP	7 (1.66)
	Others	4 (0.95)
Admission route	ED	26 (6.18)
	OD	395 (93.82)
	Referred from another medical institute	0 (0)
Severity	Mild	176 (41.81)
	Moderate	107 (25.42)
	Severe	94 (22.33)
	Extremely severe	44 (10.45)
Drug allergy records	Yes	6 (1.43)
	No	416 (98.57)
Anamnesis records	Yes	316 (75.06)
	No	105 (24.94)

Note: UEBMI: urban employee basic medical insurance, URRBMI: urban and rural resident basic medical insurance, OOP: out of pocket, ED: emergency department, *OD*: outpatient department.

**Table 3 ijerph-19-09144-t003:** Level of QoC for stroke in 2 kinds of county hospitals (*N* = 421).

QoC Indicators		All Hospitals (*N* = 421)		Comprehensive County Hospitals (*N* = 216)		TCM County Hospitals (*N* = 205)		*χ* * ^2^ *	*p **
		Recommended Care Delivered		Recommended Care Delivered		Recommended Care Delivered			
		*N*	%	*N*	%	*N*	%		
At arrival	Anticoagulant drugs	390	92.64	195	90.28	195	95.12	3.618	0.057
	Aspirin	338	80.29	170	78.70	168	81.95	0.701	0.403
At discharge	Prescriptions	301	71.50	171	79.17	130	63.41	12.806	<0.001
	Aspirin	278	66.03	156	72.22	122	59.51	7.575	0.006
	Statins	202	47.98	91	42.13	111	54.15	6.085	0.014
	Antihypertensive	85	20.19	29	13.43	56	27.32	12.595	<0.001
	Health education	5	8.93	3	16.67	2	5.26	1.953	0.314

* *p*-value tests the hypothesis that the quality of care among the 2 kinds of county hospital is the same.

**Table 4 ijerph-19-09144-t004:** Level of QoC for stroke in 3 selected counties (*N* = 421).

QoC Indicators		Linyi (*N* = 144)		Zoucheng (*N* = 144)		Qingzhou (*N* = 133)		*χ* * ^2^ *	*p **
		Recommended Care Delivered		Recommended Care Delivered		Recommended Care Delivered			
		*N*	%	*N*	%	*N*	%		
At arrival	Anticoagulant drugs	140	97.22	123	85.42	127	95.49	17.030	0.000
	Aspirin	99	68.75	118	81.94	121	90.98	21.963	<0.001
At discharge	Prescriptions	91	63.19	113	78.47	97	72.93	8.443	0.015
	Aspirin	85	59.03	102	70.83	91	68.42	4.968	0.083
	Statins	31	21.53	88	61.11	83	62.41	61.407	<0.001
	Antihypertensive	21	14.58	45	31.25	19	14.29	16.618	<0.001
	Health education	1	10.00	0	0.00	4	25.00	7.585	0.012

* *p*-value tests the hypothesis that the quality of care among 3 counties is the same.

**Table 5 ijerph-19-09144-t005:** Characteristics of the included pneumonia patients (*N* = 329).

Variables		*N* (%)
*N*		329 (100.00)
Age	0−18	75 (22.80)
	19−40	57 (17.33)
	41−60	77 (23.40)
	61−80	91 (27.66)
	>80	29 (8.81)
Gender	Male	183 (55.62)
	Female	140 (44.38)
Payment	UEBMI	67 (20.36)
	URRBMI	251 (76.29)
	Fully covered by government	0 (0.00)
	OOP	4 (1.22)
	Others	7 (2.13)
Admission route	ED	13 (3.95)
	OD	315 (95.74)
	Referred from another medical institute	1 (0.30)
Severity	Mild	93 (28.27)
	Moderate	211 (64.13)
	Severe	23 (6.99)
	Extremely severe	2 (0.61)
Drug allergy records	Yes	13 (3.95)
	No	316 (96.05)
Anamnesis records	Yes	217 (65.96)
	No	112 (34.04)

Note: UEBMI: urban employee basic medical insurance, URRBMI: urban and rural resident basic medical insurance, OOP: out of pocket, ED: emergency department, OD: outpatient department.

**Table 6 ijerph-19-09144-t006:** Level of QoC for pneumonia in 2 kinds of county hospitals (*N* = 329).

QoC Indicators		All Hospitals (*N* = 329)		Comprehensive County Hospitals (*N* = 171)		TCM County Hospitals (*N* = 170)		*χ* * ^2^ *	*p ^*^*
		Recommended Care Delivered		Recommended Care Delivered		Recommended Care Delivered			
		*N*	%	*N*	%	*N*	%		
At arrival	Blood oxygenation	112	34.04	84	50.91	28	17.07	41.937	<0.001
	Antibiotics	292	88.75	150	90.91	142	86.59	1.540	0.215
	Suggested antibiotics	298	90.58	149	90.30	149	90.85	0.029	0.864
	Blood culture	12	3.65	11	6.67	1	0.61	8.586	0.003
	Sputum culture	108	32.83	97	58.79	11	6.71	101.172	<0.001
At discharge	Health education	0	0.00	0	0.00	0	0.00	N/A	N/A

* *p*-value tests the hypothesis that the quality of care among the 2 kinds of county hospital is the same.

**Table 7 ijerph-19-09144-t007:** Level of QoC for pneumonia in 3 selected counties (*N* = 329).

QoC Indicators		Linyi (*N* = 111)		Zoucheng (*N* = 114)		Qingzhou (*N* = 116)		*χ* * ^2^ *	*p ^*^*
		Recommended Care Delivered		Recommended Care Delivered		Recommended Care Delivered			
		*N*	%	*N*	%	*N*	%		
At arrival	Blood oxygenation	3	2.52	45	43.27	64	60.38	89.343	<0.001
	Antibiotics	108	90.76	92	88.46	92	86.79	0.896	0.639
	Suggested antibiotics	103	86.55	95	91.35	100	94.34	4.086	0.130
	Blood culture	0	0.00	3	2.88	9	8.49	11.590	<0.001
	Sputum culture	47	39.50	26	25.00	35	33.02	5.291	0.071
At discharge	Health education	0	0.00	0	0.00	0	0	N/A	N/A

* *p*-value tests the hypothesis that the quality of care among 3 counties is the same.

**Table 8 ijerph-19-09144-t008:** Characteristics of the included heart failure patients (*N* = 341).

Variables		*N* (%)
*N*		341 (100.00)
Age	0–18	2 (0.59)
	19–40	8 (2.35)
	41–60	80 (23.46)
	61–80	179 (52.49
	>80	72 (21.11)
Gender	Male	219 (64.22)
	Female	122 (35.78)
Payment	UEBMI	56 (16.42)
	URRBMI	269 (78.89)
	Fully covered by government	1 (0.029)
	OOP	9 (2.64)
	Others	6 (1.76)
Admission route	ED	50 (14.66)
	OD	291 (85.34)
	Referred from another medical institute	0 (0.00)
Severity	Mild	0 (0.00)
	Moderate	277 (81.23)
	Severe	44 (12.90)
	Extremely severe	20 (5.87)
Drug allergy records	Yes	13 (3.81)
	No	328 (96.19)
Anamnesis records	Yes	304 (89.15)
	No	37 (10.85)

Note: UEBMI: urban employee basic medical insurance, URRBMI: urban and rural resident basic medical insurance, OOP: out of pocket, ED: emergency department, OD: outpatient department.

**Table 9 ijerph-19-09144-t009:** Level of QoC for heart failure in 2 kinds of county hospitals (*N* = 341).

QoC Indicators		All Hospitals (*N* = 341)		Comprehensive County Hospitals (*N* = 165)		TCM County Hospitals (*N* = 164)		*χ* * ^2^ *	*p **
		Recommended Care Delivered		Recommended Care Delivered		Recommended care delivered			
		*N*	%	*N*	%	*N*	%		
At arrival	LVS evaluation	202	59.24	112	65.50	90	52.94	5.566	0.018
	ACEI	33	9.68	17	9.94	16	9.41	0.027	0.869
	ARB	16	4.96	12	7.02	4	2.35	4.148	0.042
At discharge	Prescription	268	78.59	131	76.61	137	80.59	0.803	0.370
	Diet education	146	42.82	74	43.27	72	42.35	0.030	0.863
	Rest advice	247	72.43	129	75.44	118	69.41	1.551	0.213
	Follow-up visit advice	271	79.47	134	78.36	137	80.59	0.259	0.611

* *p*-value tests the hypothesis that the quality of care among the 2 kinds of county hospital is the same. Note: LVS: left ventricular systolic, ACEI: angiotensin converting enzyme inhibitor, ARB: angiotensin receptor blocker.

**Table 10 ijerph-19-09144-t010:** Level of QoC for heart failure in 3 selected counties (*N* = 341).

QoC Indicators		Linyi (*N* = 119)		Zoucheng (*N* = 104)		Qingzhou (*N* = 106)		*χ* * ^2^ *	*p **
		Recommended Care Delivered		Recommended Care Delivered		Recommended Care Delivered			
		*N*	%	*N*	%	*N*	%		
At arrival	LVS evaluation	58	52.25	72	63.16	72	62.07	3.354	0.187
	ACEI	8	7.21	19	16.67	6	5.17	9.839	0.007
	ARB	8	7.21	4	3.51	4	3.45	2.328	0.321
At discharge	Prescription	92	82.88	77	67.54	99	85.34	12.629	0.002
	Diet education	18	16.22	39	34.21	89	76.72	89.999	<0.001
	Rest advice	90	81.08	78	68.42	79	68.10	6.166	0.046
	Follow-up visit advice	91	81.98	79	69.30	101	87.07	11.765	0.003

* *p*-value tests the hypothesis that the quality of care among 3 counties is the same. Note: LVS: left ventricular systolic, ACEI: angiotensin converting enzyme inhibitor, ARB: angiotensin receptor blocker.

## Data Availability

The datasets generated and/or analyzed during the current study are not publicly available due further research but are available from the corresponding author on reasonable request.

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
