# Peer review of "Quality of Care in Public County Hospitals: A Cross-Sectional Study for Stroke, Pneumonia, and Heart Failure Care in Eastern China"

_ijerph, 2022, doi:10.3390/ijerph19159144_

Round 1
Reviewer 1 Report
First, thank you for allowing me to evaluate this interesting manuscript. These would be some of the considerations that I can issue on it.
I believe the introduction would benefit if some epidemiological data were included that show the reality of the three health problems selected for the work in your country. Perhaps it is not so necessary to show the peculiarities of the Chinese health system.
Table 2. Is it necessary to indicate that there is no case of stroke in people under 18 years of age? If we know that the incidence of stroke is almost non-existent in this group of diseases, what does this information provide in the table?
Line 184. Revived aspirine could be “received aspirine”?
In the discussion section, there are some paragraphs without any bibliographic reference or that seem to be mere reflections of the authors. I can understand that it is a little unvetted subject, and there is little information about it, but they should justify their claims somehow.
This is more of a question for the authors. Given what the heart failure management guidelines indicate about the use of beta-blockers, especially in people with rEF HF, do you know why has the use of this drug in its treatment not been considered a quality standard?
Author Response
Many thanks for your kind comments. Please see the attachment.

Reviewer 2 Report
This paper highlights the variance in quality of care for stroke, pneumonia and heart failure patients in China, based on internationally approved proces quality indicators for care pathways inherent to the different disease groups. Although the paper is scientifically of high value, I still have some methodological questions for the authors. Also, English language can be much improved.
Introduction:
Lines 75-77: You state that this paper focusses on the 3 most common diseases in China. Please add the prevalence in China compared to international numbers.
Methods:
Sampling:
Please highlight the year of measurement and the number of months selected. How was the random selection made?
Data extraction:
Since data was collected by taking photo's, data was manually entered in the system? Was this a double entry? How was bias reduced of controlled? Please explain.
QI's:
How was this selection made? Please add a more specific reference. How were NQF and CMA combined? Please elaborate on the choice of quality measures + definitions (numerator, denominator)
Analyses:
Please explain how quality measures were calculated. Did you investigate the variance between the different hospital scores?
Results:
Overall, the language in this section is not neutral. 'overwhelmingly', 'smaller', what does this mean? Did you compare results with international numbers? What were the expectations regarding QI scores? Please rephrase some of the sentences in this section (eg line 215, line 247).
Discussion:
I do not agree with the conclusion in lines 266 to 269 based on the results of this paper. Please compare with international numbers. Based on the data in this paper you can only make conclusions on the variety in QOC and the room for improvement based in best practice.
In line 312 you add qualitative information. How was this collected? Please add information or rephrase this sentence.
Conclusion:
'Poor level': please compare to international information. Based on the data in this paper you can only make conclusions on the variety in QOC and the room for improvement based in best practice.
Author Response
Please accept our sincerest thanks for your kind reviewing on our work. Please see the attachment.
